METHODS

# Early detection of disease outbreaks and non-outbreaks using incidence data: A framework using feature-based time series classification and machine learning

**Shan Gao**[ID][1*], **Amit K. Chakraborty**[1], **Russell Greiner**[ID][2,3], **Mark A. Lewis**[4,5‡], **Hao Wang**[ID][1‡]

**1** Department of Mathematical and Statistical Sciences, University of Alberta, Edmonton, Alberta, Canada, **2** Department of Computing Science, University of Alberta, Edmonton, Alberta, Canada, **3** Alberta Machine Intelligence Institute, Edmonton, Alberta, Canada, **4** Department of Mathematics and Statistics, University of Victoria, Victoria, British Columbia, Canada, **5** Department of Biology, University of Victoria, Victoria, British Columbia, Canada

‡ These authors jointly supervised this work.
* sgao9@ualberta.ca

## ⛉ OPEN ACCESS

**Data availability statement:** Synthetic data are available at https://doi.org/10.5281/zenodo.10967222. COVID-19 incidence data from

## Abstract

Forecasting the occurrence and absence of novel disease outbreaks is essential for disease management, yet existing methods are often context-specific, require a long preparation time, and non-outbreak prediction remains understudied. To address this gap, we propose a novel framework using a feature-based time series classification (TSC) method to forecast outbreaks and non-outbreaks. We tested our methods on synthetic data from a Susceptible–Infected–Recovered (SIR) model for slowly changing, noisy disease dynamics. Outbreak sequences give a transcritical bifurcation within a specified future time window, whereas non-outbreak (null bifurcation) sequences do not. We identified incipient differences, reflected in 22 statistical features and 5 early warning signal indicators, in time series of infectives leading to future outbreaks and non-outbreaks. Classifier performance, given by the area under the receiver-operating curve (AUC), ranged from $0.99$ for large expanding windows of training data to $0.7$ for small rolling windows. The framework is further evaluated on four empirical datasets: COVID-19 incidence data from Singapore, 18 other countries, and Edmonton, Canada, as well as SARS data from Hong Kong, with two classifiers exhibiting consistently high accuracy. Our results highlight detectable statistical features distinguishing outbreak and non-outbreak sequences well before potential occurrence, in both synthetic and real-world datasets presented in this study.

Singapore and 18 countries is sourced from Our World in Data, accessible at https://ourworldindata.org/coronavirus. The COVID-19 daily infection data comes from the City of Edmonton's Open Data Portal at https://data.edmonton.ca/Community-Services/COVID-19-in-Edmonton-Daily-Active-Cases/qkyj-dqjp. The SARS data for Hong Kong were obtained from https://www.kaggle.com/datasets/imdevskp/sars-outbreak-2003-complete-dataset?resource=download. Code to reproduce the data-processing, experiments, analysis, and figures is available at https://doi.org/10.5281/zenodo.10967222.

**Funding:** This project (S.G. and A.K.C.) was primarily supported by the One Health Modelling Network for Emerging Infections (OMNI), the Alberta Machine Intelligence Institute (Amii) matching fund, and the Department of Mathematical and Statistical Sciences (IUSEP funding) at the University of Alberta. H.W. was partially supported by the Natural Sciences and Engineering Research Council of Canada (Individual Discovery Grant RGPIN-2020-03911 and Discovery Accelerator Supplement Award RGPAS-2020-00090) and the Canada Research Chairs program (Tier 1 Canada Research Chair Award). M.A.L. gratefully acknowledges support from a Natural Sciences and Engineering Research Council of Canada (NSERC) Discovery Grant and the Gilbert and Betty Kennedy Chair in Mathematical Biology. R.G. was funded by the Natural Sciences and Engineering Research Council of Canada (NSERC, RES0043537), the Alberta Machine Intelligence Institute (Amii, RES0054194), and the Canadian Institute for Advanced Research (CIFAR, ZAEHU). The funders had no role in the study design, data collection and analysis, decision to publish, or preparation of the manuscript.

**Competing interests:** The authors have declared that no competing interests exist.

## Author summary

Timely prediction of disease outbreaks and non-outbreaks is crucial for effectively implementing preventative measures and for avoiding unnecessary overreaction. While early warning signals (EWSs) and mathematical modeling are often used for such predictions, the former may fail in systems involving stochasticity, and the latter requires underlying mechanisms that are unclear at the early stage. Here, we propose a novel framework using a feature-based time series classification method and training classifiers on these features for prediction. Within this framework, we develop a general model, with no real-world training data, that accurately forecasts outbreaks and non-outbreaks. Our results show that statistical features exhibit different distributions for outbreak and non-outbreak sequences long before outbreaks occur. We can detect these differences, evident in both synthetic and real-world datasets, to anticipate the occurrence of outbreaks and non-outbreaks. Our work would contribute to the early detection of disease outbreaks and non-outbreaks with a less sophisticated approach, providing valuable insights for mitigation strategies of novel diseases.

## Introduction

The occurrence and recurrence of infectious diseases are prevalent worldwide, with some infectious diseases showing a high mortality rate and/or strong transmissibility, while others may be nonfatal and disappear quickly without attracting much attention [1]. Balancing the risks associated with new infectious diseases and the costs of preventative measures, along with their consequences, has been a prolonged challenge. In the case of recurrent diseases like influenza, prior experience allows for informed responses, whereas we have less guidance and certainty with novel diseases. One recent example is the initial response to COVID-19, where governments started with implementing less restrictive measures, such as recommending face masks in public places, rather than enforcing mandatory quarantine policies, to minimize negative impacts on livelihoods and the economy [2–4]. Early preventative intervention is associated with lower incidence [5,6]. Anticipating infectious disease outbreaks and non-outbreaks, ideally in an early and accurate manner, therefore becomes imperative to prevent misjudgments of health risk perceptions faced by societies and their citizens, guiding the implementation of mitigation measures.

The disease outbreaks, where incidence counts escalate rapidly from a negligible level (pre-pandemic) to an uncontrollable state (pandemic) following subtle disturbances, can be viewed as critical transitions. Many natural and societal systems exhibit the potential of critical transitions, experiencing abrupt shifts from one stable state to another [7–10]. Predicting disease outbreaks (non-outbreaks) can be reframed as predicting (no) critical transition.

Many efforts have been invested in these critical transition (that is, disease outbreak) forecasts through mathematical epidemiology modeling [11,12]. These models can reflect the dynamics of infectious diseases, predict the outbreak occurrence (via the basic reproduction number $R_0$, a dimensionless value representing the expected number of secondary infections caused by a single infectious individual in a completely susceptible population [13]), estimate the long-term severity of the pandemic (using final size, representing the proportion of individuals who ultimately become infected over the disease transmission process), and hence significantly help in disease prevention and control measures. However, understanding the disease transmission mechanisms and formulating context-specific mathematical models

often necessitate sufficient data for model parameterization [14], which contradicts the limited availability of collected data at the early stage. Disease control and prevention are also time-critical endeavors. While delving into the intricate mechanisms can be data-oriented and time-consuming, prompt mitigation practices during the initial stage of a pandemic are more urgent. Mathematical modeling is better suited for mid-pandemic research and post-pandemic reflection rather than providing immediate guidance in the initial stages.

Alternatively, the early warning signal (EWS) emerges as a promising model-independent method for the early prediction of critical transitions. The fundamental principle behind the anticipation is the critical slowing down (CSD) phenomenon: a decrease in local stability when the system approaches a critical threshold, resulting in sudden shifts of alternate stable states following small perturbations [15]. Various indicators, such as increasing standard deviation [16], growing coefficient of variance [16], rising autocorrelation at lag-1 [17], changing values of skewness [18], and escalating kurtosis [19], can be used as early warning indicators (EWIs) to identify the presence of CSD.

Despite the previous success of applying EWS for predicting forthcoming catastrophic transitions [20–23], the reliability and applicability of EWS methods using the CSD as an indicator are still under debate [24,25]. The inherent stochasticity, common in disease transmission and surveillance data, can result in unforeseen dynamics and undermine the critical transition prediction. Furthermore, the smoothness assumption in EWS models may not hold in many real-world scenarios, rendering the classic "ball-in-cup" metaphor inapplicable [25]. This lack of smoothness is particularly evident in disease transmission dynamics (the pathogen evolution can be abrupt, for example). To obtain more robust and accurate predictions, we apply machine learning techniques, which have been proven effective in predicting transitions by training upon simulations [26–28].

Current endeavors on EWS theory typically concentrate on a singular object featuring critical transitions. Many investigations focus on how the changing trends in solely one indicator (or the combination of specific features) of the time series before critical transitions can be exploited [16,29,30]. In the realm of disease-related prevention and control, accurate prediction of non-outbreaks is also vital. In our context, a machine-learned classifier requires incidence time series data containing critical transitions (outbreaks) and null transitions (non-outbreaks) as the training set to capture the underlying patterns of each scenario. However, incidence data with non-outbreak are inherently scarce or usually incomplete because data collection of diseases with negligible consequences tends to be overlooked or discontinued prematurely. Considering these limitations in data availability and the need for more control over data, synthetic data can serve as an alternative option for the model training process.

We adopt the susceptible-infected-recovered (SIR) model [31] as the basis for simulating disease transmission dynamics. The SIR model, comprising susceptible, infected, and recovered individuals, can roughly reflect disease transmission dynamics in the real world. In such a deterministic model where all parameters remain the same, the epidemiological system can reach two alternate statuses: disease-free equilibrium $E_1$ if $R_0 < 1$ or disease outbreak equilibrium $E_2$ if $R_0 > 1$. Additionally, the dynamics in the real world are complex and coupled with exogenous and endogenous noise and stochasticity, unlike the deterministic model. To account for these uncertainties, we introduce three types of noise—white noise, multiplicative environmental noise, and demographic noise—to create three noise-induced SIR models for data simulation.

We simulate data exhibiting critical transition by slowly increasing the transmission rate $\beta$ and fixing other parameters. Subsequently, $R_0(t)$ becomes time-dependent and monotonically increasing. We generate 14,400 replicates of time series $I(t)$ ($t = 1, 2, ..., 1500$) from each

noise-induced SIR model, with half exhibiting a transcritical bifurcation (where $R_0(t)$ increasing through 1 for some $T \in [401, 1500]$) and the other half a null bifurcation (where $R_0(t) < 1$ for any $t \in [1, 1500]$). Finally, we define the subsequence of replicate exhibiting transcritical bifurcation $I[1 : T]$ as the pre-transition data. We label the subsequence $I[T-399 : T]$ as "T" and label the subsequence of null bifurcation data $I[t-399 : t]$ for a random $t \in [400, 1500]$ as "N" (see Fig 1B). The parameterization is justified in Chakraborty et al. [32]. Data simulation and labeling details are further illustrated in Methods.

After obtaining the simulation datasets, we train the classifier using the following features, computed from $I[1 : T]$ and $I[t-399 : t]$: (1) 22 statistical features (22SF, see Table B in S1 Text) for their outstanding performance on time series classification (TSC) tasks and minimal redundancy [33], and (2) 5 early warning signal indicators (5EWSI, see Table C in S1 Text) to represent time series data. We considered the following machine learning algorithms: (1) the gradient boosting machine (GBM), (2) the logistic regression model (LRM), (3) k-nearest neighbor (KNN), and (4) support vector machines (SVM).

Here, instead of forecasting the incidence counts or the death rate of the infectious disease, our focus lies on the early prediction of outbreak versus non-outbreak. We construct a framework for predicting pandemics using incomplete time series (i.e. $I[T-399 : T]$ and $I[t-399 : t]$) to address the challenges inherent in disease control and prevention. We train a set of classifiers using data simulated from noise-induced SIR models. We then test the classifiers on withheld simulated data to assess the performance. To investigate the adaptability, we vary the input lengths or the gap between the input and transition point (these two methods are referred to as the Rolling window and Expanding window, respectively; see Methods) and conduct the training-testing process. In the end, we test the classifiers on real-world COVID-19 incidence data and SARS data to validate their practical applicability. Performance on empirical datasets also serves as the criterion for model selection, as we tested multiple models derived from different framework configurations. Our work establishes a link between machine learning practices informed by mathematical modeling data and critical transition prediction for disease prevention and control, two disciplines that have rarely interacted cohesively before.

## Results

In the absence of prior knowledge about the novel disease, we selected multiple candidates for each configuration of the framework. As a result, we train $4 \times 2 \times 4 = 32$ classifiers, denoted as W22G-M5S (Fig 2), based on every unique combination of four simulated data sets, two feature extraction libraries, and four predictive models (see Methods). To illustrate our findings, we assess the performance of these classifiers on withheld synthetic testing sets using accuracy and the area under the receiver-operating curve (AUC) metrics.

Henceforth, we use the notation WhiteN (EnvN, DemN, or MixedN) to denote datasets comprising 14,400 replicates simulated from the SIR model with white noise (multiplicative environmental noise, demographic noise, or a mixture from other three datasets) in the following content. Sequences labeled as "T" (resp., "N") represent pre-transition subsequences of time series experiencing transcritical bifurcation – i.e., eventual outbreak (resp., subsequences exhibiting null bifurcation – i.e., normality). For simplicity, $I_{Data}$, where $Data$ is one of $W$ (WhiteN), $E$ (EnvN), $D$ (DemN), or $M$ (MixedN), represents the sliced data replicates of specified simulation. $I_{Data}^{Label}$ with $Label$ as $T$ (outbreak) or $N$ (non-outbreak), represents labeled $I_{Data}$. Additionally, 22SF (5EWSI) of $I_{Data}^{Label}$ refers to 22SF (5EWSI) computed from the specified replicates.

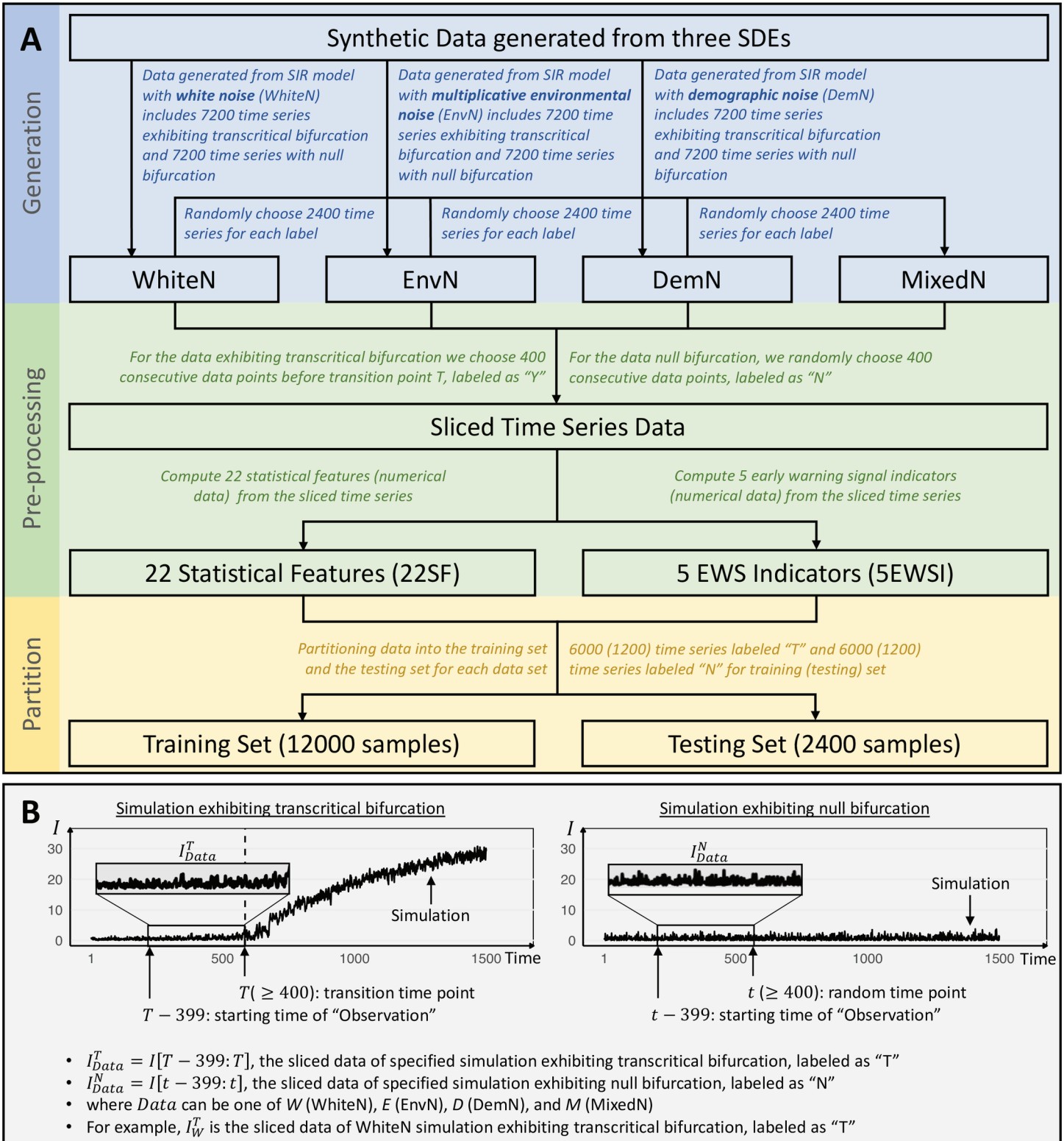

**Fig 1. Overview of the synthetic data utilized in the study.** A: The blue frame represents the data generation step, the green frame signifies data pre-processing, and the yellow frame indicates data partitioning. B: Examples of simulation from the SIR model with white noise.

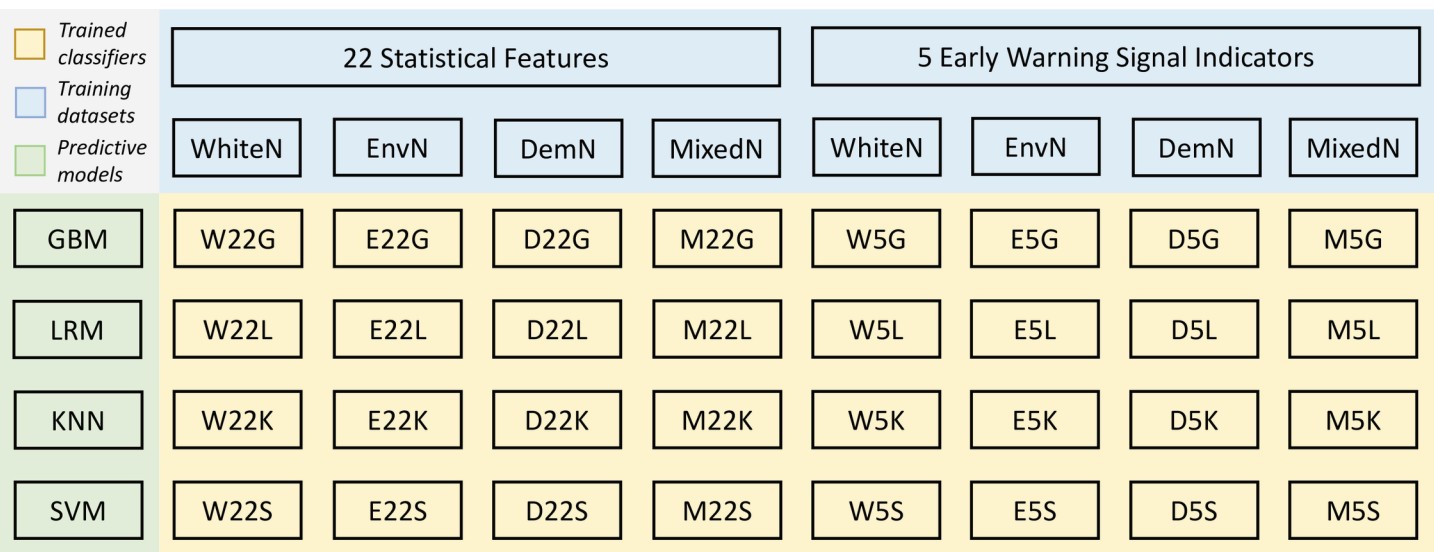

**Fig 2. 32 classifiers trained from every unique combination of four synthetic training datasets, two feature extraction libraries, and four predictive models.** The initial letter in each classifier's name corresponds to the training set (W, WhiteN; E, EnvN; D, DemN; M, MixedN). The middle number represents the feature extraction (22, 22 statistical features; 5, 5 early warning signal indicators). The last letter represents the predictive model (G, GBM; L, LRM; K, KNN; S, SVM).

## Mann-Whitney U test results on synthetic data

Features extracted from sequences with each label exhibit non-normal distribution (see Figs B-E for 22SF and Figs F-I for 5EWSI in S1 Text). Therefore, we conduct the Mann-Whitney U test, which is non-parametric and does not assume specific data distribution, on 22SF and 5EWSI computed from "T" and "N" replicates. We find statistically significant differences between features derived from "T" sequences and those from "N" sequences with $p \ll 0.001$. These findings indicate the potential to statistically discriminate between these two categories based on 22SF or 5EWSI.

## Performance on withheld testing set

We evaluate the classification performance of synthetic-data-trained classifiers on withheld testing sets, as presented in Fig 3 (detailed data are provided in Table D in S1 Text). Results show that all the classifiers can achieve near-perfect performance on withheld testing sets, with AUC scores ranging from E5K ($0.9911 \pm 0.0038$) to D22G, D22K, D22L, D22S, D5G, D5K, D5L, D5S, W22L, W22S, W5G, and W5L ($1 \pm 0.0000$). Given the high AUC scores, we generated t-distributed stochastic neighbour embedding (t-SNE) plots (Fig J in S1 Text), which visually confirm the clear distinction between the two classes, supporting the near-perfect classification performance on the withheld testing set.

The AUC scores of 32 different classifiers are consistently high, exceeding 0.9900 and being numerically close. Hence we conduct DeLong tests [34] to compare the AUC scores of different classifiers. Tables A-D in S2 Text present p-values of the DeLong tests across eight training sets with a predetermined model (GBM, LRM, KNN, and SVM, respectively). Classifiers trained using 5EWSI of $I_E$ (i.e., E5G, E5L, E5K, or E5S) generally exhibit slightly lower AUC scores compared to classifiers trained using the same predictive model but with different datasets ($p < 0.001$ for most classifiers).

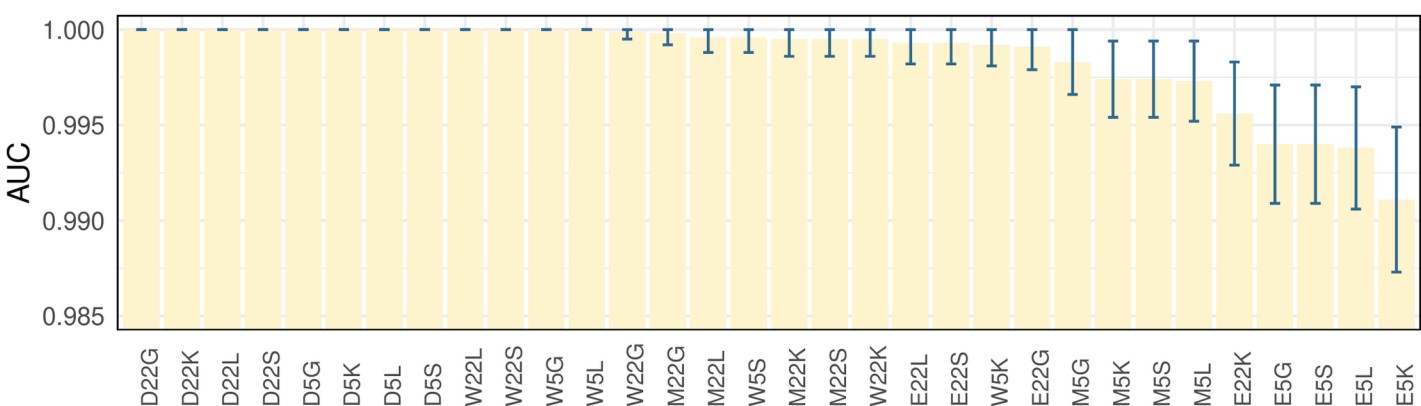

**Fig 3. The AUC values of 32 synthetic-data-trained classifiers (horizontal axis, see Fig 2) on withheld testing sets.** Classifiers are reordered by AUC scores. Error bars correspond to the 95% confidence intervals. DeLong tests are conducted to compare the AUC values of classifiers, with detailed results available in Tables A-D in S2 Text (where the predictive model is fixed) and Tables E-L in S2 Text (where the training data is fixed).

Tables E-L in S2 Text present the DeLong test results among four models, trained on the same dataset. Counterintuitively, despite the distinct properties of various predictive models and their ability to handle different types of data, the test results suggest no significant difference in AUC scores among four predictive models when applied to most of the synthetic testing sets. However, there are some exceptions: when using 22SF of $I_E$ (see Table F in S2 Text), the AUC of four classifiers, E22K has the lowest AUC score ($0.9956 \pm 0.0027$) compared to E22S ($0.9993 \pm 0.0011$, $p < 0.01$), E22L ($0.9993 \pm 0.0011$, $p < 0.01$), or E22G ($0.9991 \pm 0.0012$, $p < 0.01$). Moreover, as illustrated in Fig 3, classifiers trained by 5EWSI of $I_E$ across four models rank the last four in terms of AUC scores, with E5K ($0.9911 \pm 0.0038$, $p < 0.05$) being the lowest, while the AUC scores among E5G, E5S, and E5L are statistically similar ($p > 0.05$). This means 5EWSI of $I_E$ is the most challenging one to be distinguished among the four models.

### Performance on withheld testing set with varying sequence length and gap

In this context, replicates $I_{Data}^T$ end at the transition points, leaving no response time for implementing mitigation measures. Despite the satisfactory performance of 32 classifiers over withheld testing sets, two questions naturally emerge: (1) How early can we predict an impending disease outbreak? (2) How much data is required for prediction to maintain a certain level of accuracy? To address these concerns, we conduct two experiments using the Rolling window (where the distance between data and the transition point varies while fixing the data length) and the Expanding window (where the data length varies while the distance between data and the transition point is fixed), as illustrated in Figs 4A and 5A. We repeatedly conduct training-testing processes using Rolling windows or Expanding windows, resulting in corresponding classifiers at each round. The AUC values and classification accuracy are depicted by points using different colors and shapes. Results are shown in Figs 4 and 5 for AUC and Fig L in S1 Text for accuracy.

The results of the Rolling window (Figs 4B–4I) illustrate that approaching the transition point can generally increase the AUC scores for all classifiers, among which those trained from the GBM model showing better performance across four types of noise data. Classifiers trained on 5EWSI of any noise data can achieve AUC scores of almost 1 when the gap is

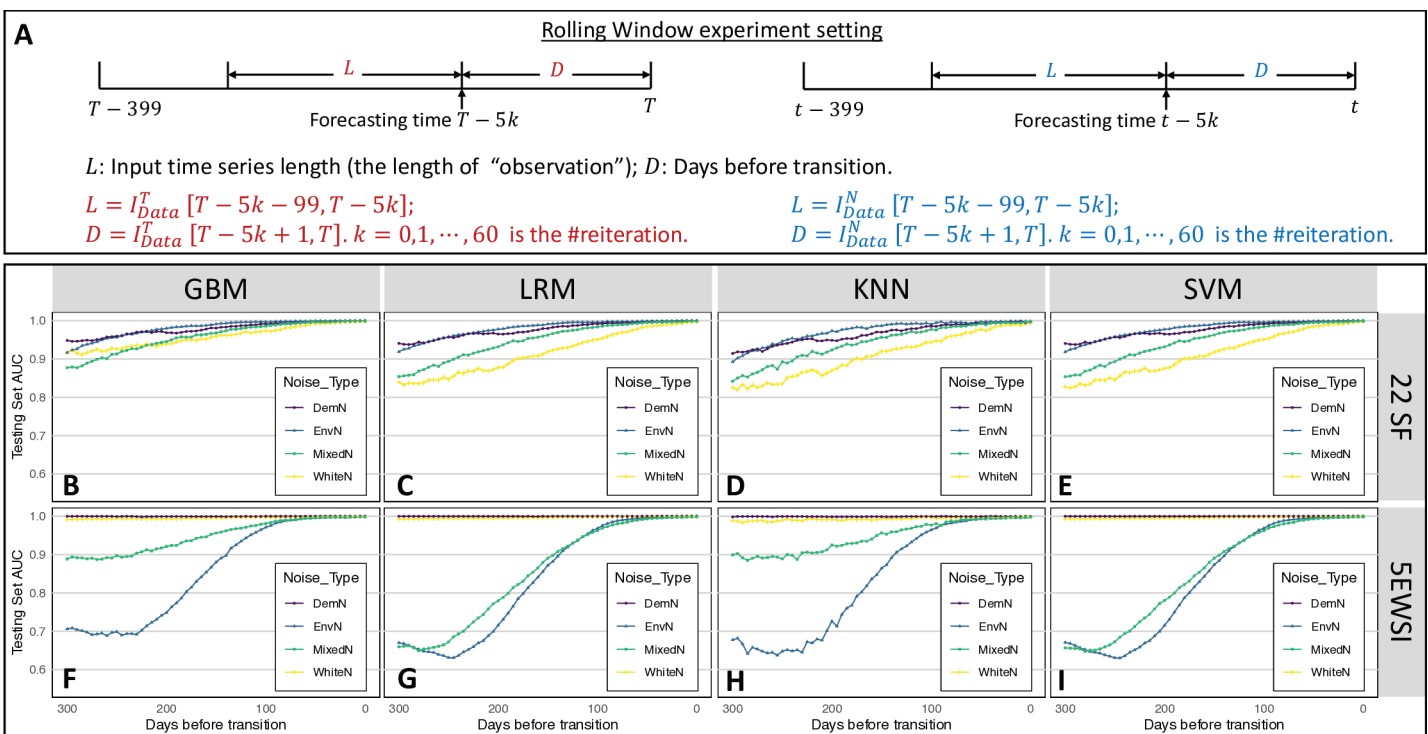

**Fig 4. Illustration in Rolling window experiment settings and AUC score at different reiterations.** A: Rolling window settings on $I_{Data}^T$ and $I_{Data}^N$. We maintain a fixed length of input time series $L$ and roll the window $L$ from the right to the left boundary. B-I: AUC scores depicting the classification performance of the classifier trained from changing gaps $D$ between the subsequence and transition points. Two feature extraction methods (22SF, 22 statistical features; 5EWSI, 5 early warning signal indicators) are put in two rows and four predictive models (GBM, gradient boosting machine; LRM, logistic regression model; KNN, k-nearest neighbor; SVM, support vector machine) are presented in four columns.

shorter than 50, while those employing 22SF require a distance of less than 30 to reach similar performance as 5EWSI. Classifiers trained on 22SF of $I_E$ and $I_D$ slightly outperform those trained on the other two noise data for all four models (see blue triangle and purple round points in Figs 4B–4E. However, 5EWSI of $I_E$ and $I_M$ exhibit notably lower AUC across predictive models when the distance is greater than 100 (see blue triangle and green square points in Figs 4F–4I. We also observe that when the distance between the time series to the transition point is less than 100, two feature extractions computed from any of the sliced time series exhibit similarly excellent performance across four models. As the window moves farther from the transition point, at a distance of 300 data points for example, the GBM model can achieve better performance for both $I_D$ and $I_M$ in terms of feature extraction. We further notice that classifiers trained from 5EWSI of $I_W$ or $I_D$ can achieve better and more robust performance as their AUC values do not increase substantially along with closer distances (see yellow cross and purple round points in Figs 4F–4I. In contrast, classifiers trained from 5EWSI of $I_D$ and $I_M$ yield significantly improved classification performance when far from the transition points (see green square and blue triangle points in Figs 4F–4I). Additionally, it is worth noting that the results of LRM and SVM models present negligible differences (less than 0.001 in general).

The results of the Expanding window experiments are presented in Figs 5B–5I. AUC scores exhibit a rapid improvement as we increase the length of the testing time series from 5 to 100, preserving an AUC score of over 0.97 thereafter, which suggests that the classifiers derived

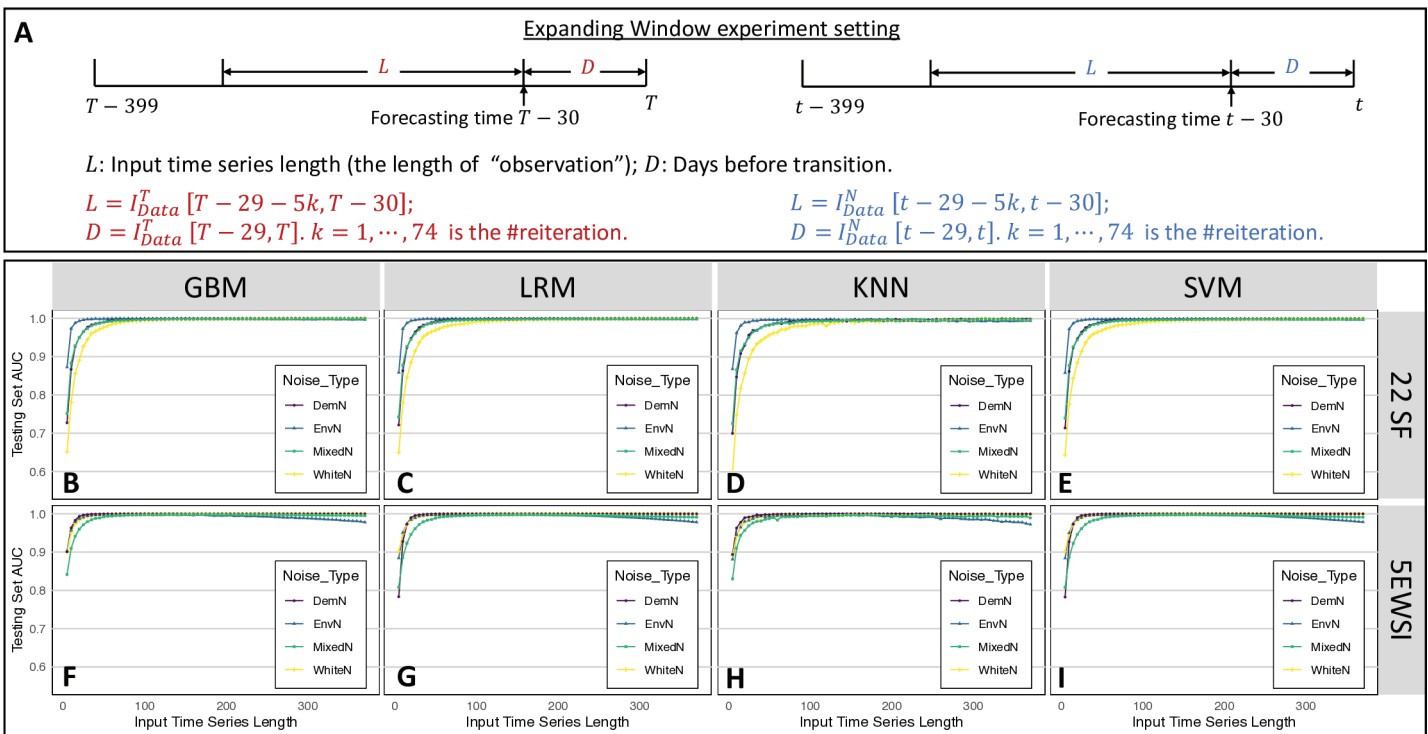

**Fig 5. Illustration in Expanding window experiment settings and AUC score at different reiterations.** A: Expanding window settings on $I_{Data}^T$ and $I_{Data}^N$. We fix the length of $D$ as 30, then expand $L$ until reaching the left boundary of sliced data. The initial length of $L$ is 5. B-I: AUC scores depicting the classification performance of the classifier trained from changing gaps $D$ between the subsequence and transition points. Two feature extraction methods (22SF, 22 statistical features; 5EWSI, 5 early warning signal indicators) are put in two rows and four predictive models (GBM, gradient boosting machine; LRM, logistic regression model; KNN, k-nearest neighbor; SVM, support vector machine) are presented in four columns.

from the framework can remain applicable even for short synthetic time series. While longer time series data intuitively contribute to better performance, we observe a slight decreasing trend in AUC for classifiers trained by 5EWSI when the input time series length approaches 370, likely due to redundant data distant from the transition point. Interestingly, classifiers trained from 22SF of the original time series show no such decline in AUC. One speculation is that classifiers trained on data represented by 22SF are more robust when dealing with data redundancy.

## Performance on empirical testing set

Although our models perform well on withheld test sets, we need to further evaluate their practical utility using real-world datasets. Performance on empirical datasets is also the criterion for model selection as it provides insight into the optimal choices for the model of simulation, feature extraction, and the machine learning algorithm within the framework.

We tested the performance of 32 simulation-trained classifiers (see Fig 2) on four empirical datasets: two at the national level, COVID-19 data from Singapore ($I_{SG}$) and from 18 countries ($I_{18}$), and two at the city level, COVID-19 data from Edmonton, Canada ($I_{ED}$), and SARS data from Hong Kong ($I_{HK}$). Although violating the assumption of homogeneous population distribution in the SIR model, we utilize $I_{18}$ to evaluate the performance of classifiers on data that deviates from the model assumption. We use the $I_{ED}$ to compare the performance with existing methods. Note that three datasets ($I_{SG}$, $I_{18}$, and $I_{HK}$) solely contain replicates labeled

as either "T" or "N", making it impossible to compute AUC scores. As a result, we use the classification accuracy as the performance measurement on testing sets, and the results are presented in Fig 6.

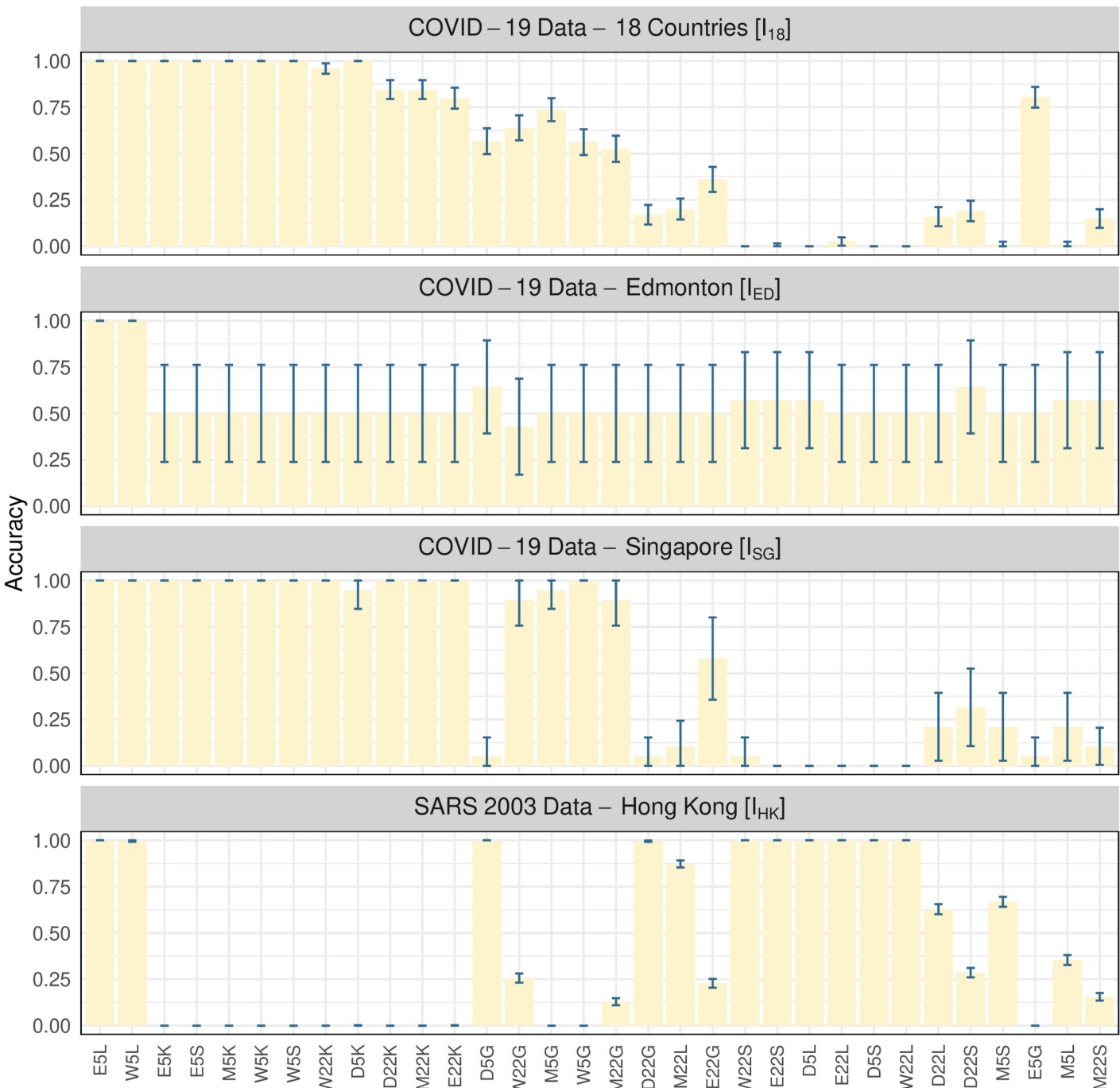

**Fig 6. Classification accuracy of 32 synthetic-data-trained classifiers (horizontal axis, see Fig 2) to correctly classify subsequences of four empirical datasets.** Error bars correspond to the 95% confidence intervals.

As shown in Fig 6, the performance varies across the four empirical datasets depending on the type of data used for model training, specifically the simulations generated from the SIR model with different types of noise. Notably, there are two models, *E5L* and *W5L*, that consistently demonstrate high classification accuracy across all empirical datasets, making them the best-performing models among the 32 candidates. These findings suggest that simulating the SIR model with either white noise or environmental noise is preferable. Additionally, computing five classic early warning indicators from the data and using the logistic regression model would provide better classification performance.

For dataset $I_{SG}$, 17 out of 32 models performed better than random with 16 achieving an accuracy of 0.9 or higher. For dataset $I_{18}$, 18 models achieved an accuracy above 0.5, but only 9 models exceeded 0.9. Although $I_{SG}$ and $I_{18}$ are both national-level datasets, we observed generally lower accuracy on $I_{18}$ compared to $I_{SG}$, which is consistent with our rationale that Singapore data is more suitable for SIR model assumptions. The performance of $I_{18}$, $I_{SG}$ and $I_{HK}$, or in other words, between replicates solely labeled as "T" (outbreaks) or "N" (non-outbreaks) seem to be complementary. Most classifiers performing well in classifying "T" replicates fail to predict replicates labeled as "N", except for two classifiers, E5L and W5L, achieving an accuracy of 1 ($\pm 0.0000$) in out-of-sample empirical data classification tasks. For dataset $I_{ED}$, which is also used in Chakraborty et al. [32] as the empirical testing set, the best two models (AUC = 1) outperformed the models trained by classic early warning signals (AUC < 0.5), the deep learning model proposed by Bury et al. [26] (AUC < 0.5), and Chakraborty et al. [32] model (AUC = 0.71). Note that the error bar for $I_{ED}$ is significantly larger compared to the other three datasets as its sample size is much smaller.

Although the minimum data length of $I_{SG}$ and $I_{18}$ is 14, we further access the performance of the classifiers on shorter replicates by removing 7 data points (i.e., ending the data one week before the outbreak timing) from these two COVID-19 datasets. Results presented in Tables M and N in S1 Text show that if the disease outbreak occurs seven days later, there is no significant difference between using the current data for prediction and additionally incorporating data from the next seven days. Thus, we can effectively predict the outbreak with a lead time of one week. It is worth noting that the "N" replicates are selected from the subset of SARS data with $R_e < 1$, with random lengths, hence we do not conduct the same test on shorter replicates for this dataset. We should also acknowledge that asymptomatic or presymptomatic individuals exhibit scant influence on flu transmission [35], in contrast, COVID-19 incidence data are significantly underestimated due to these "unnoticeable" infections. We expand the empirical data on individuals infected with COVID-19 five-fold and repeat the experiment. The results in Tables O and P in S1 Text show that underestimation of incidence data does not have a significant impact on classification performance, no matter whether the original performance is excellent or undesirable.

## Discussion

Timely predictions of disease outbreaks can expedite the decision-making process and thus minimize associated losses. That said, implementing proactive and stringent actions when forecasting impending disease outbreaks; or promoting conservative and relatively mild policies when predicting non-outbreaks. Nevertheless, current efforts in early predictions of disease outbreaks and non-outbreaks, while urgent, are still far from perfect.

In our work, we introduce a novel framework for early forecasting the outbreaks (replicates labeled as "T") and non-outbreaks (replicates labeled as "N") using feature-based time series classification approaches. We train 32 classifiers following the framework, all achieving near-perfect classification performance. Further experiments show that these classifiers can handle

time series of varying lengths and those far from the transition point (that is, outbreak timing). Notably, two classifiers, trained from 5EWSI of sliced time series and logistic regression model, exhibit consistently good performance over four empirical testing sets that are never used in the training process. These results may suggest that predicting impending transitions, or disease outbreaks in the context of epidemiology, can be achieved by a less sophisticated method.

Early prediction of transitions is not new. Numerous novel EWIs, as well as their combinations, are proposed for transition prediction. Tredennick et al. [22] detect CSD using EWIs in the context of measles transmission in Niger; Harris et al. [20] demonstrate that several statistical features exhibited specific characteristics before the outbreak of malaria in Kericho, Kenya; O'Brien and Clements [21] access the reliability of EWSs through daily COVID-19 incidence sequences of 24 countries. Moreover, Bury et al. [26] apply deep learning algorithms to classify different types of bifurcations using simulated data. However, disease-related applications of EWS have been conducted with outbreak cases, while cases where non-outbreaks occur are overlooked. In addition to achieving excellent performance in the detection of impending pandemics, we also aim to undermine the misjudgment of normality cases, i.e. reduce the type 2 error. In this study, we simulated time series data undergoing transcritical bifurcation (outbreak) and null bifurcation (non-outbreak) and achieved a classification accuracy exceeding 0.99 in these simulations.

The limited length of time series, which is common in incidence data, is also a lingering problem for early prediction of outbreaks [21,36]. We should further consider the trade-off between earliness and accuracy. Note that we can manipulate the length and location of the subsequences of the raw time series. With replicates farther from the outbreak timing, we observe consistently less performative results for classifiers except those trained from $I_W$ or $I_D$ of 5EWSI. This result is consistent with the CSD phenomenon that inner stability decreases and data fluctuation becomes more pronounced when approaching the transition point, resulting in the difference with those far from the transition point. Data will be more informative when closer to the transition points. For extremely short training data, particularly those with lengths less than 20, classifiers exhibit lower AUC scores due to insufficient information learned from feature extraction. Still, they outperform the random chances. As the length increases, the AUC increases rapidly at first and then tends to stabilize. Similar to our work, Nanopoulos et al. [37] apply four statistical features (the mean, standard deviation, skewness, and kurtosis) of the time series to conduct the TSC tasks, and claim that feature-based methods are more robust compared to valued-based ones.

Delecroix et al. [38] retrieve studies on infectious disease outbreaks anticipation via EWIs, and find that more than half are conducted on simulated data only. It is premature to conclude the practicability purely based on simulations. In addition to applying synthetic data generated from stochastic differential equations (SDE) models, we assess the performance of classifiers on empirical datasets to predict whether outbreaks will occur. Although the classifiers are not trained on empirical datasets, some classifiers trained from synthetic data can still make correct classifications, as illustrated in Fig 6.

It is noteworthy that the performance of the models for the two types of data seems to be complementary, even though the training sets are balanced. The other finding in performance is that 5EWSI, with lower dimensionality than 22SF, outperforms 22SF on real-world testing data, despite both sets of features in this paper achieving near-perfect classification accuracy and AUC scores of approximately 1 on synthetic data. This discrepancy in performance may be attributed to model overfitting using 22SF. Moreover, classifiers trained from the KNN model demonstrate outstanding performance over withheld testing sets from synthetic and

empirical data labeled as "T" (COVID-19 data from Singapore $I_{SG}$ and 18 countries $I_{18}$). However, these classifiers consistently failed to classify "N" data (SARS data from Hong Kong $I_{HK}$). The KNN model classifies the new objective based on its "distance" from the training set, essentially storing instances of the training set without constructing a model. This may also elucidate why classifiers derived from KNN achieved good performance on withheld testing sets: for the KNN model, empirical "N" data is "closer" to simulated "T" data.

Generic EWS predictive ability is limited [24,25,39], but identifying mechanistic behavior [40] and incorporating specific information can enhance the prediction performance. Thomson et al. [41] show the efficacy of early warning systems for malaria-outbreak anticipation using meteorological, epidemiological, and environmental factors. Climate- and weather-driven early warning systems can handle potential disease threats [42]. Our goal, however, is to make early predictions by assuming no prior knowledge of the infectious disease which is an unavoidable fact when a novel disease emerges. Classifiers derived from our proposed framework only require incidence data, the most straightforward data from surveillance.

There are several limitations in our work. The SIR model used in this work operates under the assumption of homogeneous mixing of the infected and susceptible populations. Such a prerequisite does not hold for large and complex communities (countries, for instance). During the simulation process, we assume a linear increase in the transmission rate over time. Future research will aim to address these limitations by exploring more realistic non-linear functions for the transmission rate. Furthermore, our work does not address two widespread concerns: (1) when the pandemic will emerge after the onset of the disease, and (2) how severe the potential pandemic could lead to without control policies. With an appropriate mathematical model, the former question can be addressed through effective reproduction number ($R_e$) estimation, while the latter can be theoretically answered by determining the final epidemic size. However, the dilemma lies in the fact that rigorous mathematical modeling itself requires a relatively longer time, and the assumptions of such the modeling process are counting on biological and epidemiological studies on the unknown disease. As a potential direction for future research, developing a more effective predictive approach on both outbreaks and non-outbreaks requires greater attention to address these gaps. Finally, while simulations are used to enrich the training data for the models, inherent discrepancies exist between simulated and real-world data, potentially impacting the model's performance in practical scenarios. We therefore incorporate uncertainties into the simulation process by introducing various types of noise into the SIR model and randomizing the outbreak emergence times. Future research could further conduct theoretical analysis to examine how the noise and parameters can affect the classification outcomes using uncertainty quantification (UQ) [43].

To conclude, our study reveals the inherent statistical differences in time series exhibiting outbreaks and non-outbreaks before their occurrence. Such discrepancies, in turn, enable the classification of these two scenarios proceeding the outbreak or non-outbreak happens. Thus, we introduce a framework for early predictions of disease outbreaks using the feature-based TSC method, and results demonstrate that all classifiers can accurately predict the outbreaks and non-outbreaks in a simulated context, with two achieving an accuracy of 1 on out-of-sample empirical data. Our work presents a less sophisticated yet effective approach to predicting disease outbreaks and non-outbreaks, addressing the realistic requirements of early disease emergence stages.

## Methods

### Time series classification and predictive models

For a univariate time series $I(t)$, feature-based methods perform a feature extraction from $I(t)$ before the classification procedure. The computed features serve to transform the time series into a structured format comprising $N$ numeric variables, allowing us to directly employ machine learning algorithms to classify the feature data, and subsequently accomplish TSC tasks. Here, we employ four predictive models, namely GBM, LRM, KNN, and SVM, for classification use. Among these, GBM and KNN are non-linear classifier models, LRM is a linear model, and a non-linear kernel was chosen for SVM. We additionally considered the simple decision tree (SDT) as it is more interpretable than GBM. However, SDT-trained models perform less effectively on withheld data than GBM-trained models, we therefore focus on the GBM. Detailed results of SDT, as well as the importance of features, are provided in Table E and Fig K in S1 Text. AUC scores [44] of classifiers and classification accuracy on testing sets are recorded as performance measures.

In this work, the code and analysis are carried out using the R. Two models, GBM and KNN, are implemented using the `caret` package [45]. The SVM model is applied using the `e1071` package [46]. The LRM model, which is the logistic regression model in our work, is computed by the `LRM` function contained in the `stats` package of R [47]. The AUC of each classifier is calculated using the `pROC` package [48].

### Synthetic data used for training and testing

We first elaborate on how the synthetic data are simulated from dynamic models. Then we explain how the simulated time series are pre-processed for classification use. Note that synthetic data are used for both the training and testing processes in this paper.

**Synthetic time series generation.** We start the investigation on the classic susceptible-infectious-recovered (SIR) model [31]:

$$\begin{cases} \frac{dS(t)}{dt} = \Lambda - \beta(t)S(t)I(t) - \mu S(t), \\ \frac{dI(t)}{dt} = \beta(t)S(t)I(t) - \alpha I(t) - \mu I(t), \\ \frac{dR(t)}{dt} = \alpha I(t) - \mu R(t), \end{cases} \tag{1}$$

where $S$, $I$, and $R$ are susceptible, infected, and recovered individuals at time $t$, respectively. $\Lambda$ is the recruitment rate of susceptible population, $\beta(t)$ is the transmission rate at time $t$, $\mu$ is the death rate, and $\alpha$ is the recovery rate.

Admittedly, the SIR model is often regarded as simplistic in capturing the dynamics of infectious diseases. More complex and context-specific models are better suited for simulating the intricate transmission mechanisms in real-world scenarios. But it is also the simplicity of the SIR model that provides the "greatest common factor" to accommodate the novel disease to some extent. Furthermore, the dynamics of infectious diseases can be easily influenced by random internal and/or external disturbances. To augment the SIR model, we followed Chakraborty et al. [32] and introduced three distinct stochastic differential equations (SDEs) incorporating white noise, multiplicative environmental noise, and demographic noise to generate synthetic incidence time series. The rationale behind selecting these specific noise types, as well as the parametrization, are justified in Chakraborty et al.[32].

Mathematically, the basic reproductive number $R_0$, a dimensionless value representing the expected count of secondary infections caused by a single infectious individual in a completely susceptible population, serves as an important metric for the stability of equilibrium

points in the SIR model [49]. When $R_0 < 1$, the disease-free equilibrium $E_1 = (\Lambda, 0, 0)$ is stable, and the system remains in a non-outbreak state. $R_0 > 1$ indicates the potential for the disease to persist within the population, and the endemic equilibrium $E_2 = (\frac{\mu+\alpha}{\beta}, \frac{\Lambda}{\mu+\alpha} - \frac{\mu}{\beta}, \frac{\alpha\Lambda}{\mu(\mu+\alpha)} - \frac{\alpha}{\beta})$ becomes stable. At the critical threshold of $R_0 = 1$, a transcritical bifurcation occurs, wherein the disease-free equilibrium and endemic equilibrium intersect and exchange their stability properties. Following the methodology proposed by Chakraborty et al. [32], all parameters except the transmission rate $\beta$ are to be constants. Then the value of $R_0$ becomes solely dependent on $\beta(t)$, expressed as $R_0(t) = K\beta(t)$, with $K = \frac{\Lambda}{\mu(\alpha+\mu)}$. We further assume a linear variation in the transmission rate $\beta$ with respect to time $t$: $\beta(t) = \beta_0 + \beta_1 t$, where $\beta_0$ and $\beta_1$ follow triangular distributions. Starting with a random initial value of $\beta(t = 0)$ such that $R_0(t = 0) < 1$, we gradually increase $\beta$ over time to make $R_0$ exceed 1 to obtain the outbreak simulation. During this process, the presence of noise ensures that the infected individual $I$ increases after the critical point. Without noise, $I$ would become and remain zero when $R_0$ is initially less than 1. If the slope $\beta_1$ is sufficiently small, not exceeding the critical value of reproduction number within the time interval $t \in [1, 1500]$, the simulation results in null bifurcation.

**Synthetic data input. WhiteN (EnvN, DemN, or MixedN) (# replicates = 14,400; Data simulated from SIR model with noise)** Following the simulation process outlined in section Synthetic Time Series Generation, we generate three datasets from three corresponding SDEs, denoted as WhiteN (SIR model with white noise), EnvN (SIR model with multiplicative environmental noise), and DemN (SIR model with demographic noise). For each SDE model, 7,200 replicates $I[1 : 1500]$ (univariate time series) exhibiting transcritical bifurcation are generated. Note that replicates with transition time points $T$ smaller than 400 are excluded to avoid inconsistencies in the following data-slicing process. An additional 7,200 replicates $I[1 : 1500]$ where no bifurcation event occurs are also generated from each SDE model. Simulations for both scenarios from each SDE model are visualized in Fig A in S1 Text. Furthermore, from each category in the three datasets, we randomly select 2,400 replicates, forming a new dataset named MixedN.

Since we aim to predict the outbreak before its occurrence, it is imperative to exclude the data after the transition point $T$ for full-time series $I[1 : 1500]$ exhibiting transcritical bifurcation when forming the training sets and testing sets. Therefore, we extract 400 data points before the transition point and label it as "T", as illustrated in Fig 1B. To maintain consistency, 400 consecutive data points are randomly chosen for each simulation with null bifurcation. A summary of the synthetic data used in this study is described in Fig 1.

## Empirical data used for testing

To assess the applicability of the framework in real scenarios, we apply empirical time series data of two infectious diseases: the novel coronavirus disease (COVID-19) and severe acute respiratory syndrome (SARS). COVID-19 has emerged as an excellent subject for novel disease studies due to the abundance of accessible data, both temporally and spatially, and its prolonged duration. COVID-19 exemplifies novel infectious diseases causing significant outbreaks (transcritical bifurcation). The SARS outbreak was initially identified in November 2002 [50]. Since 2004, there have been no known cases of SARS reported worldwide, providing an illustrative example of non-outbreak (null bifurcation) in our study.

Similar to synthetic data, we apply the effective reproduction number ($R_e$) as the criterion to determine whether empirical time series exhibiting outbreaks (transcritical bifurcation) and the timing of outbreak occurrence. Given a full-time series $I$ exhibiting outbreak(s), $I[k_0 : k]$ is labeled as "T" if and only if $R_e[t] < 1$ for any $k_0 \leq t \leq k$, $R_e[k_0 - 1] \geq 1$, and $R_e[k + 1] \geq 1$. For $I$ exhibiting the non-outbreak, any subsets will be labeled as "N".

$I_{SG}$ (**# replicates = 19; COVID-19 daily infection data from Singapore**) The fundamental assumption of the classic SIR model is the homogeneity of population distribution (i.e. each individual in the population has an equal contact probability). However, this assumption may not hold for data spanning a geographically extensive area, such as national-level data, despite their potential for greater accuracy and availability. Nonetheless, the COVID-19 data from Singapore presents an ideal subject for study, where the population demographics align well with the assumptions of the SIR model. Additionally, the government responded promptly to the pandemic, implementing intensive tracking via mobile technology (Government Technology Agency of Singapore [51]), enhancing data accuracy. To maintain the consistency of the SIR model and the empirical testing set, we focus on the COVID-19 infection data from Singapore, spanning from January 26, 2020, to February 18, 2024. The data is collected weekly and then smoothed by evenly distributing the infection number over seven days of the corresponding week to obtain daily data. $R_e$ is estimated via `R` package `EpiEstim` [52] with a mean value of 6.3 and a standard deviation of 4.2 for the serial interval [53]. Replicates with lengths shorter than 14 days are excluded, resulting in a testing set comprising 19 replicates labeled as "T".

$I_{ED}$ (**# replicates = 14; COVID-19 daily infection data from Edmonton, Canada**) The dataset $I_{ED}$ was obtained from the City of Edmonton's Open Data Portal [54] and includes daily incidence data from March 2020 to July 2023. Similar to $I_{SG}$, the $R_0$ was estimated using `R` package `EpiEstim` [52], and subsequences with $R_0 < 0$ and length is larger than 56 (8 weeks) were selected. Note that this empirical dataset and the data processing are identical to the empirical datasets used in [32] for comparison purposes.

$I_{18}$ (**# replicates = 194; COVID-19 incidence data from 18 countries**) While the national data are often underestimated (due to presymptomatic and asymptomatic case) and against the homogeneity of the population, we still use the national level data to test the robustness of the model. To derive the replicates and the corresponding labels, we apply the $R_e$ estimation from Huisman et al. [55] for COVID-19 incidence data, where the assumption is that the incidence data come from symptomatic individuals. Given the typical lag between disease occurrence and data collection, subsequences with lengths shorter than 14 days (i.e., two weeks), or replicates containing missing data are excluded. The final dataset comprises 194 replicates, labeled as "T", encompassing data from 18 countries. Detailed information on the sliced data is provided in Table H in S1 Text.

$I_{HK}$ (**# replicates = 1200; SARS 2003 incidence data from Hong Kong**) We utilize SARS data from Hong Kong, spanning from March 17 to July 11, 2003 [56]. The original data comprises Cumulative Cases, Death Cases, and Recovered Cases. To maintain consistency with the SIR model, we computed the prevalence (i.e., the "I" term in the SIR model) by subtracting "Death Cases" and "Recovered" from Cumulative Cases. The recorded data displays missing values, with one missing data point every Sunday for the initial 11 weeks and two missing data points every Saturday and Sunday in the last 5 weeks (see Fig O in S1 Text). To impute the missing data, we applied `R` package `imputeTS` using `na_interpolation` and obtained the completed incidence time series. The mean and standard deviation of the serial interval are 8.4 and 3.8, respectively [57]. We found that the mean value of estimated $R_e$ falls below 1 during the estimation interval from April 18 to April 24, 2003, and never exceeds 1 afterward. To ensure greater confidence in the $R_e$ values being less than 1 for the subsequences, we designate the sequence after a later date, May 15, 2003, as the null bifurcation scenario and randomly select 1200 subsets, each with a random length larger than 7, as the "N" replicates in the testing set.

## Statistical features and early warning signal indicators

Classifying raw time series, especially those observed at higher frequencies, usually involves working with high-dimensional data, which can be computationally intensive and less interpretable [58]. To address this concern, people seek a proper representation of the original time series, typically encompassing statistical attributes, in a numerical format. In this paper, we employ two distinct sets of features, 22SF and 5EWSI, to represent the sequence for classification purposes.

Numerous alternative features for time series are available, and among them, 22SF computed through R package `Rcatch22` stands out as a promising choice, as it exhibits ideal performance in TSC and minimal redundancy [59]. The 22SF utilized in this study are detailed in Table B in S1 Text, and more information is available in Lubba et al. [33]. Note that the 22SF of the simulated time series do not follow normal distribution, therefore, we conducted the Mann-Whitney U test on these features computed from "T" and "N" sequences across four types of noise data. The results are depicted in Figs B-E in S1 Text. As can be easily observed, the features computed from data with two labels are significantly different, with $p \ll 0.001$. These findings suggest that despite the visually indistinguishable nature of $I_{Data}^{T}$ and $I_{Data}^{N}$, these two types of sequences can be theoretically separated by examining these statistical features.

Alternatively, we consider 5 early warning signal indicators (5EWSI) from time series to serve as the features, with their formulas provided in Table C in S1 Text. Notably, there is no overlapping between 22SF and 5EWSI. We also implemented the Mann-Whitney U test, and results demonstrate that there is a difference in these five features computed from time series labeled as "T" and "N" (Figs F-I in S1 Text).

## Experiment settings for earliness and accuracy

Maximizing classifier accuracy using the complete time series is typically the primary goal of TSC problems [60]. However, in time-sensitive applications, such as infectious disease monitoring and pandemic forecasting, the aim shifts to optimizing the balance between two contradictory (or contradicting or opposing) objectives: accuracy and earliness in the classification task. Such transition implies that the classifier should ideally handle inadequate (shorter time intervals) and early (those far from the impending transition) sequences while maintaining a certain level of classification accuracy to provide reliable references for policymakers. Consequently, critical questions arise in two-fold: How early can predictions be reliably provided? What is the sufficient length of testing time series required to offer a reliable prediction? To address these questions, we conducted two experiments, as illustrated in Figs 4A and 5A.

Firstly, we employ the Rolling window approach on both $I_{Data}^{T}$ and $I_{Data}^{N}$. We maintain a fixed length of 100 for input time series $L$ and "roll" the $L$ from right to left, resulting in an increasing gap $D$ between the last time point of the window and the transition point $T$. Starting with an initial gap of 0, we increment the gap by 5 units for each iteration $k$, yielding 61 experiments. After determining the input time series $L$ for each round, we follow the same feature-computation and data partition procedure outlined in section Synthetic Time Series Generation and train classifiers through four models. We record the AUC and classification accuracy on testing sets for each classifier at every iteration.

To further investigate predictive performance using varying lengths of the time series data, we conduct training and testing processes using an Expanding window approach. Initiating with a 5-time-point input time series $L$, we expand the $L$ by adding five more data points to the left end of the time series for each iteration, while maintaining the length of gap $D$ as 30.

Consequently, the length of input time series $L$ ranges from 5 to 370, resulting in 74 reiterations for each time series dataset and each model. We again follow the feature extraction, and training-testing sets partition approaches, and perform the classification. We repeat these experiments across all eight datasets and four models, recording the corresponding AUC scores and accuracy values for assessments.

## Supporting information

**S1 Text. The supplementary text, tables, and figures.**
(PDF)

**S2 Text. The supplementary text and tables for DeLong test.**
(PDF)

## Acknowledgments

We thank Pouria Ramazi, Tianyu Guan, Reza Miry, Ilhem Bouderbala, and Russell Milne for their valuable feedback.

## Author contributions

**Conceptualization:** Shan Gao, Mark A Lewis, Hao Wang.

**Data curation:** Shan Gao, Amit K Chakraborty.

**Formal analysis:** Shan Gao.

**Funding acquisition:** Russell Greiner, Mark A Lewis, Hao Wang.

**Investigation:** Shan Gao.

**Methodology:** Shan Gao, Mark A Lewis, Hao Wang.

**Project administration:** Mark A Lewis, Hao Wang.

**Software:** Shan Gao, Amit K Chakraborty.

**Supervision:** Mark A Lewis, Hao Wang.

**Visualization:** Shan Gao.

**Writing – original draft:** Shan Gao.

**Writing – review & editing:** Shan Gao, Amit K Chakraborty, Russell Greiner, Mark A Lewis, Hao Wang.

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
