## [Decision Letter · Decision Letter 0]

4 Oct 2024

Dear PhD Student Gao,

Thank you very much for submitting your manuscript "Early detection of disease outbreaks and non-outbreaks using incidence data: A framework using feature-based time series classification and machine learning" for consideration at PLOS Computational Biology.

As with all papers reviewed by the journal, your manuscript was reviewed by members of the editorial board and by several independent reviewers. In light of the reviews (below this email), we would like to invite the resubmission of a significantly-revised version that takes into account the reviewers' comments.

Thanks for the submission! Both reviewers acknowledged that combining machine learning (ML) and dynamical systems approaches to better forecast epidemics is an interesting approach. However, they raised major concerns with the work. Please address the reviewers' comments especially the ones regarding the uncertainty quantification of model forecasts (in my opinion, a very relevant issue in practical use of the model) and the reliability of the model trained on synthetic datasets (i.e. a common problem for ML models due to lack of large real-world datasets).

We cannot make any decision about publication until we have seen the revised manuscript and your response to the reviewers' comments. Your revised manuscript is also likely to be sent to reviewers for further evaluation.

Sincerely,

Ruian Ke

Guest Editor

PLOS Computational Biology

Thomas Leitner

Section Editor

PLOS Computational Biology

Thanks for the submission! Both reviewers acknowledged that combining machine learning (ML) and dynamical systems approaches to better forecast epidemics is an interesting approach. However, they raised major concerns with the work. Please address the reviewers' comments especially the ones regarding the uncertainty quantification of model forecasts (in my opinion, a very relevant issue in practical use of the model) and the reliability of the model trained on synthetic datasets (i.e. a common problem for ML models due to lack of large real-world datasets).

Reviewer's Responses to Questions

**Comments to the Authors:**

Reviewer #1: The authors propose a machine learning framework to predict disease outbreak. They train machine learning classifiers on simulation data from a stochastic SIR model. They generate a bifurcation class, where the model is taken through a transcritical bifurcation, and a null class, where the model does not pass through a bifurcation. The classifiers are trained on the binary classification problem of distinguishing these classes. They perform with very high accuracy on a withheld simulation test set. When evaluated on real data from Covid and SARS, 2 out of the 32 trained models perform with high accuracy.

This paper is written well, has clean figures and a good, logical structure. The subject of the study—combining machine learning and dynamical systems to predict disease outbreaks—is interesting and timely. The combinatorial analysis of the different classifiers/noise/features is thorough, as well as the investigation with respect to rolling vs. expanding windows. Overall, the analysis of the approach using model simulations is well done.

However, I am not convinced that this approach, in its current state, is effective on real data. Figure 4 shows that the majority of the trained classifiers perform no better than random (an accuracy of around 0.5 when combining the empirical datasets to balance the classes). This would suggest that the simulated training set is not sufficiently representative of what is occurring in the real data. The fact that 2 out of the 32 models perform well is not a sufficient indication that generalizable features have been learned from the training set to predict real disease outbreak—-These models may have scored high by chance, given the number of models tested. I would need to see further evidence of the performance of these two classifiers on real data before agreeing with the final statement of the abstract---that there are “statistical features distinguishing outbreak and non-outbreak…in real-world datasets”.

On that note, why not do the Mann-Whitney tests on the features obtained from the real data? Do you observe significant differences between outbreak vs. non-outbreak? Are they as one would expect based on dynamical systems theory? If not, maybe that is the issue here.

Personally, I find the analysis on the synthetic data interesting in itself, and a step forward in developing generalized machine learning approaches for predicting bifurcations. This alone could be an interesting publication in a more specialised journal. If the authors believe that this approach is effective on real data as it stands, then I will need more convincing.

Reviewer #2: The work presented in this manuscript tests the hypothesis that a machine learning model based on features extracted from time series data can be used to forecast whether a transition point in an epidemic will occur. In general, this is an interesting approach which uses machine learning models trained on synthetic data and applies them to real world data. However, my main critique explained in several points below is that the authors train a wide array of models based on choices of features and the machine learning method, but it’s not clear which model should be used in practice. Uncertainty quantification would provide statistical confidence in the predictions, but this is missing from the study.

1) Given that the AUC scores are close to 1, this seems to strongly imply separability of the different classes based on the input features to the machine learning models. If this is the case, it would be useful to visualize this using dimensionality reductions techniques down to 2 or 3 dimensions, e.g., using PCA or t-SNE.

2) I think it would be helpful to plot the noiseless data with the noisy data, e.g., in Figure 1, to visualize how much noise is added to the data.

3) A comparison to noiseless data for the results in Figure 3 would be able to show whether adding noise affects the classification accuracy. I think Figure 3 shows that in some cases, because the AUC is exactly equal to 1, Demographic or White noise doesn’t reduce classification accuracy.

4) Figure 6 shows that the results between classifiers on real world data are heterogeneous. In practice, I don’t think it would be possible to know a priori which classifier would be better to use. Can the authors discuss how can one choose the best classifier?

5) Given that some classifiers achieve 1.0 accuracy, it would be informative if the authors showed the results of training a linear classifier model. In this case, the coefficients of the linear model could be used to interpret which features are driving the high accuracy. Similarly, the authors can report the results of a simple decision tree instead of a gradient boosting model, since the decision tree model will be more interpretable.

6) Can the authors provide uncertainty quantification for the machine learning model predictions? If not, how can one be confident in the prediction?

7) Can the authors provide an explanation/interpretation for why the different types of noise yield different results in figure 7?

8) Model E5L has one of the lowest AUCs in figure 3, and hence one would assume it wouldn’t perform well on real data. However, it has the highest accuracy in figure 6 when testing on real data. How can this be explained?

**Have the authors made all data and (if applicable) computational code underlying the findings in their manuscript fully available?**

Reviewer #1: Yes

Reviewer #2: Yes

PLOS authors have the option to publish the peer review history of their article (what does this mean?). If published, this will include your full peer review and any attached files.

Reviewer #1: No

Reviewer #2: No
---

## [Decision Letter · Decision Letter 1]

10 Jan 2025

Dear Gao,

We are pleased to inform you that your manuscript 'Early detection of disease outbreaks and non-outbreaks using incidence data: A framework using feature-based time series classification and machine learning' has been provisionally accepted for publication in PLOS Computational Biology.

Best regards,

Ruian Ke

Guest Editor

PLOS Computational Biology

Thomas Leitner

Section Editor

PLOS Computational Biology

Reviewer's Responses to Questions

**Comments to the Authors:**

Reviewer #1: The authors have addressed my concerns by evaluating their best-performing models on other disease data. I'm happy to recommend publication.

Reviewer #2: Thanks to the authors for addressing all of my previous concerns and comments.

**Have the authors made all data and (if applicable) computational code underlying the findings in their manuscript fully available?**

Reviewer #1: Yes

Reviewer #2: Yes

PLOS authors have the option to publish the peer review history of their article (what does this mean?). If published, this will include your full peer review and any attached files.

Reviewer #1: No

Reviewer #2: No

---

## [Editor Report · Acceptance letter]

PCOMPBIOL-D-24-00668R1

Early detection of disease outbreaks and non-outbreaks using incidence data: A framework using feature-based time series classification and machine learning

Dear Dr Gao,

I am pleased to inform you that your manuscript has been formally accepted for publication in PLOS Computational Biology. Your manuscript is now with our production department and you will be notified of the publication date in due course.

With kind regards,

Anita Estes
